# Role of *Helicobacter pylori* Infection in Pathogenesis, Evolution, and Complication of Atherosclerotic Plaque

**DOI:** 10.3390/biomedicines12020400

**Published:** 2024-02-08

**Authors:** Tiziana Ciarambino, Pietro Crispino, Giovanni Minervini, Mauro Giordano

**Affiliations:** 1Internal Medicine Department, Hospital of Marcianise, ASL Caserta, 81037 Caserta, Italy; 2Internal Medicine Department, Hospital of Latina, ASL Latina, 04100 Latina, Italy; pcri@libero.it; 3Internal Medicine Department, Hospital of Lagonegro, AOR San Carlo, 85042 Lagonegro, Italy; gminer@libero.it; 4Department of Advanced Medical and Surgical Sciences, University of Campania “L. Vanvitelli”, 81100 Naples, Italy; mauro.giordano@unicampania.it

**Keywords:** *Helicobacter pylori*, pathogenesis, atherosclerotic plaque

## Abstract

The therapeutic management of atherosclerosis focuses almost exclusively on the reduction of plasma cholesterol levels. An important role in the genesis and evolution of atherosclerosis is played by chronic inflammation in promoting thrombosis phenomena after atheroma rupture. This review aims to take stock of the knowledge so far accumulated on the role of endemic HP infection in atherosclerosis. The studies produced so far have demonstrated a causal relationship between Helicobacter pylori (HP) and CVD. In a previous study, we demonstrated in HP-positive patients that thrombin and plasma fragment 1 + 2 production was proportionally related to tumor necrosis factor-alpha levels and that eradication of the infection resulted in a reduction of inflammation. At the end of our review, we can state that HP slightly affects the risk of CVD, particularly if the infection is associated with cytotoxic damage, and HP screening could have a clinically significant role in patients with a high risk of CVD. Considering the high prevalence of HP infection, an infection screening could be of great clinical utility in patients at high risk of CVD.

## 1. Introduction

Cardiovascular disease (CVD) caused by atherosclerosis ranks first among the most common and most prevalent diseases in the world [1]. At the basis of this pathology, there is the atherosclerotic process characterized by a long latency period which involves alterations of glucose metabolism, inflammatory and immune factors, alterations of the delicate balance existing between the endothelial barrier, and finally, the cascade activation of the blood coagulation system (Figure 1) [2]. Furthermore, the role of unconventional risk factors, such as that played by the microbiome, has recently been observed, indicating that the composition of the intestinal bacterial flora can play a causal role in the formation, progression, and complication of atherosclerotic lesions [2]. The pathological culmination of this extremely long and insidious process is the degeneration and fissuring of the atherosclerotic plaque, which clinically manifests itself with a more or less severe acute ischemia of vital organs, which has as its price an increase in disability and mortality in the short and middle terms. Cardiovascular events and atherosclerosis have different fates among various individuals, also diversifying according to gender. Despite that CVD are caused by numerous factors such as lifestyle, nutrition, and voluptuous habits of individuals, it is known that these factors alone cannot fully explain the dynamics present in these diseases by suggesting the existence of generically determined elements that act as constitutive risk factors, which in some cases are gender specific [3,4]. The development process of atherosclerosis is based on metabolic changes that gradually start from the age of 20 and concern the molecular processing of lipids, although in the presence of metabolic abnormalities such as dyslipidemia, diabetes, and obesity, they are also commonly observed in childhood and adolescence, manifesting themselves with small subclinical atherosclerotic lesions. The main organic factor of this disease is the excessive deposition of lipids in the intima of the arterial walls, which determines the formation in the arteries of fibro-lipid plaques, and subsequently, all the dramatic upheavals that concern the remodeling and degeneration of the plaque itself. As has long been known from the data available in the literature [5,6], vascular events originate from the fissure of the plaque, which leads to an anomalous activation of the haemo-coagulative cascade and the development of thrombosis [7].

A fundamental role in the process of atherogenesis is played by the vascular endothelium through the synthesis of nitric oxide. Nitric oxide (NO) is an important determinant of systemic circulation and actively contributes to modulating processes such as the inhibition of platelet adhesion and aggregation and the maintenance of physiological pressure inside the arteries [8]. NO therefore hinders the development of atherosclerotic plaque [9] and minimizes the proliferation and increase in thickness of vascular smooth muscle [10]. From the data available in the literature, it is recognized that inflammation is associated with an increased risk of cardiovascular events, regardless of cholesterol levels circulating in the blood [11,12]. In order to corroborate this relationship, some plasma inflammatory markers such as highly sensitive C-reactive protein (hs-CRP), adhesion molecules such as the type 1 soluble intercellular adhesion molecule (sICAM-1), serum amyloid A, various cytokines such as interleukin-6 (IL-6) and interleukin-8 (IL-8) have been tested as predictors of CVD risk, although the role remains to be defined.

In the last twenty years, numerous pieces of scientific evidence have correlated the chronic systemic inflammatory state with the presence of various infectious pathogens, in particular with HP [13,14,15,16]. HP is a Gram-negative bacterium, with a microaerophilic metabolism able to damage the mucosal barrier of the stomach, causing gastritis, peptic ulcer, and eventually gastric cancer [17]. Growing evidence has shown that HP infection is associated with extra gastric manifestations such as CVD, but other studies do not find this association significant [18,19,20]. Since there are conflicting opinions in the literature on the relationship between HP infection, chronic inflammatory state, and atherosclerosis, we believe that a more in-depth study of the evidence is needed. The purpose of our review is to shed light on the role of the inflammatory state induced by HP in atherosclerosis in order to establish whether the identification of this infection has its usefulness and above all, to shed light on the basis of evidence in the literature such as the population groups to be screened.

## 2. Methods

We performed a narrative review, carrying out a search of the literature in PubMed related to the topic, identifying the studies published up to December 31, 2022. In order to search for the works most relevant to the topic concerning the association between HP and atherosclerosis and cardiovascular risk, we used the following search terms in the titles and abstracts: (‘*Helicobacter pylori*’ or ‘*H. pylori*’ or ‘*Helicobacter*’ or ‘*Campylobacter pylori*’ or ‘helicobacter infections’) and (‘chronic systemic inflammation’ or ‘myo-intimal thickening’ or ‘intima-media thickness’ or ‘carotid atherosclerosis’ or ‘intracerebral atherosclerosis’ or ‘systemic atherosclerosis’ or ‘silent atherosclerosis’) or ‘coronary atherosclerosis’ or ‘hypercoagulability’. We took into consideration data from the results of experimental studies, while comments, case reports, or studies considered repetitive and similar in sample or content were excluded. Data extraction and quality assessment of the papers were conducted by two reviewers (PC and TC) who independently extracted data and assessed the quality of each study. For each pooled article, we collected the first author’s name, publication date, study design, study country, HP detection methods, age, male ratio, and the number of participants. Any discrepancies of opinion between reviewers in research selection, quality assessment, or data extraction were addressed, subsequently widening the assessment to the other two authors (GM and OP). The supervision of all the work carried out was carried out by MG, the creator and coordinator of the research.

## 3. Pathogenesis of Atherosclerosis: The Centrality of Endothelial Integrity

Atherosclerosis originates from toxic damage to the vascular endothelium linked to the accumulation of lipids associated with a persistent inflammatory state that alters the production of nitric oxide [20,21,22,23,24,25] (Figure 1). Nitric oxide (NO) regulates the tone of smooth muscle cells and of the vascular wall in response to a stimulus that induces a vasodilatory action [25,26]. The endothelium is also an active tissue from a metabolic point of view because it is able to produce other molecules with a vasogenic function such as prostaglandin I2 and the hyperpolarizing factor derived from the endothelium, and molecules with a vasoconstrictor function including endothelin-1, thromboxane A2, and angiotensin II [27,28]. It is precisely the perfect balance between these molecules that maintains endothelial integrity by preventing the oxidative, thrombotic, and inflammatory damage underlying CVD. These pathologies depend on the decrease in NO, on the decrease in the turnover of vascular smooth muscle cells, and finally, on the loss of the endothelial capacity to prevent leukocyte adhesion and platelet aggregation and adhesion [29]. The sum of these processes increases vascular permeability by modifying the blood flow speed and favors the passage of elements harmful to the organism (bacteria, toxins, etc.) [30].

### 3.1. Immune Function of Endothelial Cells and Atherosclerosis

The mechanisms regulating blood flow and the immune function of the endothelium are intimately interrelated [29]. Endothelial cells express immunologically active immune system receptors such as Toll-like receptor (TLR) and Nod-like receptor (NLR) proteins [31,32]. Toll-like endothelial receptor 4 (TLR4) appears to influence blood flow based on the degree of inflammation present [33]. In confirmation of what has been said, a constant and turbulence-free blood flow makes arterial endothelial cells insensitive to the passage of ligands such as the Toll-like receptor (TLR2), lipopolysaccharide (LPS), and tumor necrosis factor (TNF) [33].

Furthermore, healthy endothelial cells are capable of producing cytokines and chemokines in response to a foreign antigenic stimulus and interleukins with pro-inflammatory activity (IL-8) through the stimulation of the NOD1/NF-kB system, which is activated following stimulation by microorganisms [29].

Endothelial cells can promote immune function and T lymphocyte recruitment by acting as antigen-presenting elements as well as the class II major histocompatibility complex (MCHC-II) [34].

Endothelial cells are capable of engulfing bacteria through the opsonization of their particles and complementary activation [35,36]. Endothelial cells are also involved in the apoptosis of neutrophils by inducing their phagocytosis under the influence of interleukin-1 [37] and in the presence of arachidonic acid in the production of antimicrobial peptides [38,39,40].

### 3.2. Inflammation of Endothelial Cells and Atherosclerosis

The loss of the normal function of the endothelium largely linked to the decrease in the synthesis of nitric oxide (NO) represents the initial moment in the evolution of the atherosclerotic process. Subsequently, this is associated with inflammation of the vascular wall, induced by the deposition of lipoproteins [29], to the point of attracting leukocytes and platelets to adhere to the lesion [41]. The decrease in NO and above all, the decrease in eNOS (metabolically active) with respect to the iNOS isoform, increases the intraparietal inflammation favored by the release of pro-inflammatory cytokines [29,42]. iNOS impairs endothelial function and limits the activity of prostacyclin synthase, compromising the vasodilator function of the endothelium [43,44]. Proper NO production contributes to low iNOS levels through the inhibition of nuclear-factor-activated B-cell (NF-B) signaling, which promotes transcription and production of the iNOS isoform [45].

Another mechanism relating to NO and inflammation in the vascular wall involves the participation of some lipid mediators. For example, the enzyme phospholipase A2 (PLA2) induces the release of arachidonic acid by degrading plasma membrane phospholipids, thus providing a substrate capable of triggering the inflammatory cascade [45]. As we have seen, NO plays a role in the inflammatory cascade, which assumes arachidonic acid as its main substrate. This action is related to the enhancement of cyclooxygenase (COX) activity, which, by promoting the production of prostaglandins, induces inflammation [46,47,48]. Some data believe that the amino acid L-arginine is essential for the production of NO, thus also favoring the regulation of the immune response [49,50]. T lymphocytes but above all, macrophages use arginine in a differentiated way according to their function [49,50].

Moreover, it would seem that the enhancement of the enzymatic activity of arginases at the level of endothelial cells is directly proportional to the entity of the inflammatory state and to the senescence of the endothelium as well as to the vascular reflux modeling [51]. Prostaglandin-2 (PGE2) can also modulate inflammatory activity and at the same time contribute to blood flow [52], as well as leukotrienes capable of inhibiting the activity of NF-κB and cell adhesion molecules (VCAM-1, ICAM-1), E-selectin, and monocyte chemoattractant protein-1 (MCP-1) [53,54].

Endothelin-1 (ET-1) is a vasoconstrictor peptide secreted in response to stimulation by TNF-α, IL-1, and IL-6 [29,55,56].

### 3.3. Platelets Aggregation and Clotting Formation in Atherosclerosis

It is known that platelets are unable to adhere to the vascular wall and aggregate with each other in the presence of an intact endothelium and without flow turbulence [57]. In response to a pro-aggregating stimulus, or under the influence of thromboxane A2 (TxA2) and adenosine diphosphate (ADP) response, platelets can attract other cell types and continue to aggregate [58], activating glycoprotein IIb/IIIa), which makes it related to coagulation factors, and in the form of integrin αIIbβ3 by binding to fibrinogen and von Willebrand factor (VWF) [59,60,61]. The participation of platelets in atherosclerosis does not only occur acutely during the rupture of an atherosclerotic plaque but also during the genesis of the plaque [59,62] by interacting with the macrophages present at the level of the lipid core, contributing to the enlargement of the fibro-lipidic layer [63,64,65]. Furthermore, they interact with monocytes causing a functional transformation that has characteristics similar to those of myofibroblasts, thus contributing to the development of the fibrous cap in the context of atherosclerotic plaque [66]. During cap rupture, monocyte–platelet complexes play an active role in promoting inflammation through the production of cytokines, with a regulatory and chemotactic action towards T lymphocytes and natural killer cells [67,68,69,70]. This linking activity between inflammatory and thrombotic processes is due to platelet factor 4 (PF4), CD40, and migration inhibitory factor (MIF) [71,72], under the effect of platelet-activating factor (PAF) produced from endothelial cells, monocytes, and platelets themselves. PAF binds to its receptor (PAFR) inducing its aggregation [73].

## 4. *H. pylori* and Pathogenicity

HP is an extremely common pathogen in the world population that causes chronic infection and a consequent persistent inflammatory state characterized by local damage to the gastric mucosa but which also has systemic relevance as it is able to constantly stimulate the resident immune cells causing the release of cytokines, which have characteristics similar to those involved in the processes of atherosclerosis [74,75,76].

The degree of bacterial virulence and the characteristics of the host are very relevant in making HP infection generate pathological events [77]. In particular, the genetic phenotype of the host likely influences the progression of clinical manifestations associated with HP infection. In fact, specific polymorphisms in the genes coding for some cytokines (IL-1β, IL-8, IL-10, and TNF-α) cause a more severe infection and a greater inflammatory response, as well as being associated with an increased risk of gastric cancer [78]. Furthermore, the polymorphisms concerning the genes that regulate the innate immune processes, such as those concerning the Toll-like receptor 4 (TLR4), are of great importance because the extent of the epithelial damage of the gastric mucosa depends on their function [78].

### 4.1. Etiopathogenesis of H. pylori Colonization

The development of HP infection is mainly due to the conformation of the bacterium, the flexibility of the cytoskeleton, and the flagellar movement and occurs in three consecutive stages:adhesion and colonization of the gastric mucosa,evasion of the immune response,induction of mucosal damage.

The tropism for the gastric mucosa is variable and depends, as we have said, on the characteristics of the host. In fact, once the stomach has been colonized, HP reaches the epithelium via some receptors defined as Tlp, which, by chemotaxis, direct the flagellar movement in response to precise chemical signals (urea, lactate, and reactive oxygen species) present in the gastric environment [79,80,81]. Due to the effect of the neutral pH induced by ammonia, there is also a reduction in the viscosity of the mucus, which allows the bacteria to move freely [80,81]. HP adhesion occurs mainly thanks to the affinity of the lipopolysaccharides present on the HP surface with Lewis antigens, surface glycoproteins of the gastric epithelium capable of modulating cell–cell adhesion [82,83,84,85]. Among the proteins of the bacterial wall of HP, we find the blood antigen A binding protein (BabA) and the sialic acid binding protein (SabA), which promote adhesion to the gastric cell by binding, respectively, to host LeB and LeX [86,87]. SabA, once bound to its receptor on the gastric wall, also stimulates an immune response by neutrophil granulocytes, thus modulating the intensity of the inflammatory response [88,89].

The adhesion of HP to the gastric surface activates the protein complex (T4SS), which allows the translocation of cytotoxins including group A-linked cytotoxin (CagA) onto gastric epithelial cells. This phenomenon silences host immune recruitment and thus HP is able to evade the antigenic response to non-self [90,91]. Another cytotoxin, VacA, induces apoptosis of the host cell favoring the formation of vacuoles, which constitute a protective niche in which HP can lodge in the shelter of the immune response [92]. Expression of the toxin VacA is common to all strains of HP, while the vacuolating cytotoxin (CagA) is usually expressed only by some strains which are associated with the most severe inflammatory manifestations [91,93].

### 4.2. Role of Inflammation in H. pylori Infection: From Gastric Mucosa to Systemic Translocation

Much data has emerged on the correlation between inflammation induced by HP infection and pathological conditions characterized by a chronic inflammatory infiltrate such as atherosclerosis [94]. The development of systemic diseases related to HP infection largely depends on the extent of inflammation triggered by the adaptive immune response although studies involving the two main virulence factors have not led to conclusive results [95]. This would suggest that the systemic complications of HP depend not only on the virulence of the strain but also on how the host immune defenses react to the infection and on concomitant external factors such as smoking, diet, and environmental factors [95]. The inflammatory response triggered by HP is widely variable and can cause local and systemic release of proinflammatory cytokines, or in some cases, some strains that have matured lower free-radical-releasing metabolic systems induce a milder inflammatory response [96].

In other words, depending on their metabolism, some strains of HP can modulate the antigenic response and therefore modulate the extent of the inflammatory response [97,98,99]. This is largely due to the fact that some strains of HP are able to use superoxide dismutase (SOD) and therefore reduce oxidative stress and with it suppress the production of proinflammatory cytokines induced by its presence [100], while others instead are able to trigger a greater inflammatory response, with the release of cytokines (IL-1β and IL-18) [101]. Furthermore, it would appear that HP is able to manipulate the host microbiome by modulating hydrochloride-peptic secretion and influencing the composition of the gastric mucosal barrier [102,103], interfering with the expression and production of mucins [104]. Interestingly, HP is not only able to influence the integrity of the gastric barrier and the composition of the local microbiome but also the composition of the bacterial flora resident in other systems distant from the gastrointestinal one [78]. For this reason, it is assumed that HP can determine pathological manifestations in sites distant from the stomach through the translocation of regulatory T cells (Treg) and immunologically active molecules which, in the presence of an altered permeability of the intestinal barrier, are able to regulate immune tolerance [105]. This phenomenon could contribute to the progression of extra-gastric diseases.

## 5. *H. pylori* and Atherosclerosis

We have said that the role of HP in extra gastric pathologies is variable, multifactorial, and not always obvious. One study demonstrated that HP infection affects the ability to release vascular endothelium and consequently, eradication of the infection improved endothelium-dependent vasodilation [106]. According to He J. et al. [107], it is possible to summarize the main hypotheses supporting the role of HP in determining systemic pathologies. Therefore, it is possible to hypothesize that HP induces a systemic inflammatory reaction because:It induces the production of various pro-inflammatory factors (TNF-α, IFN-γ, IL-1, IL-2, IL-6, IL-8, IL-1).It indirectly causes the formation of immune complexes that mimic cross-antigenic reactions, common to autoimmune pathological manifestations.

These are the various mechanisms by which HP would appear to play a role in inducing atherosclerosis. Alongside the mechanisms of cytokine activation described up to now, at least two other hypotheses of endothelial damage by HP have been identified. First of all, it has been observed that the infection associated with CagA positivity is more associated with these diseases [108]. The mechanisms through which CagA would be due to the increase in the production of COX-1 and COX-2 at the level of the vascular endothelium, thereby stimulating the release of prostaglandins and thromboxane A2 (TXA2) by prompting platelets to aggregate [108]. It is also possible that an aberrant immune reaction, such as that exerted by CagA antibodies on the antigens of the endothelial wall, is a determinant element for the fissure of the fibro-lipidic plaque [107,108].

Myocardial infarction is the most serious manifestation of atherosclerotic processes. In this regard, a meta-analysis observed that among the risk factors for heart attack among young subjects, there was positivity to HP infection [109]. Also, with regard to the risk of atherothrombotic stroke, an epidemiological study demonstrated that HP positivity was associated with a greater risk of carotid atherosclerosis in Asian subjects under the age of 50 [110]. If HP has been directly associated with the progression of carotid atherosclerosis, another study has also demonstrated the role of the bacterium in metabolic syndrome and how it can also indirectly affect the formation and degeneration of carotid plaques [111]. As has already been reported, not all scientific evidence correlates HP infection with atherosclerosis, and therefore in the next sections, we will analyze the evidence for and against this hypothesis.

### 5.1. Experimental Evidence in Favor of the Role of H. pylori in Atherosclerosis

We have, from a statistical point of view and some studies involving many participants, observed that HP infection correlates significantly with myocardial infarction and atherothrombotic stroke, but we have not yet mentioned the direct evidence of this association [109,110].

In this section, we will discuss the direct evidence that has led to the suspicion of HP as one of the determining factors in the progression of atherosclerotic manifestations (Figure 2). Already in the 1990s, it was postulated that these chronic infections, such as those from HP, were associated with a persistent state capable of increasing the concentrations of molecules of the acute phase such as fibrinogen and sialic acid [112].

Based on this data, Patel et al. [112] showed that in the group of patients with ischemic heart disease, more than 1/3 were seropositive for HP compared to the normal population, in the absence of differences in socioeconomic status and risk factors for cardiovascular disease. Indeed, high fibrinogen concentrations and total leukocyte counts were independently present in HP seropositive men. However, the same author suggested that the data was not conclusive and that this relationship could be conditioned by true confounding factors, including the presence of a non-specific low-grade chronic inflammatory response not strictly dependent on HP. In 1997, Markle et al. [113] hypothesized that there was a potential causal relationship between cardiovascular disease and the quality variation in the intraluminal acid content present in conjunction with HP-induced gastritis, indicating reduced folate bioavailability. The folate reduction in turn led to an inhibition of methionine synthase, increasing the blood concentration of homocysteine, known to be another important risk factor for atherosclerosis and endothelial damage. Birnie et al. [114] highlight an increase in anti-heat shock proteins, a group of acute-phase molecules, in patients with HP infection and coronary heart disease, hypothesizing that bacterial exposure leads to a significantly increased risk of coronary heart disease by means of an autoimmune process. Pascerì et al. [115] observed that strains of HP carrying cytotoxin-associated gene A (CagA) were more common in patients with coronary heart disease than in normal individuals for equal risk factors. Therefore, the hypothesis that HP may influence atherogenesis through prolonged low-threshold systemic inflammation is considered valid. Ameriso et al. [116] found the presence of HP by means of immunohistochemical reaction methods in atherosclerotic localizations of the carotid artery deriving from subjects undergoing endarterectomy.

In addition to the direct presence of HP, an immunohistochemical investigation of the intercellular adhesion molecule-1 (ICAM-1), a marker related to the inflammatory cellular response, was performed. The direct presence of the microorganism in the atherosclerotic context was a common characteristic of the male sex and was independent of the extent of the neurological symptoms and of the general characteristics of the individuals. The authors concluded that HP is associated with the inflammatory cellular response with evidence for the relationship between infection and atherosclerotic disease. The present study did not ascertain the mechanisms by which HP is able to colonize carotid lesions, nor whether the bacterium is transiently or permanently present within the atherosclerotic plaque [116].

The same study highlighted that there are at least three different scenarios.

First, it has been hypothesized that acute infection may precipitate ischemic events, especially in individuals with a significant presence of vascular risk factors. The key to this phenomenon is due to the transient imbalance of the blood coagulation system towards a prothrombotic state.Secondly, HP infection causes a chronic inflammatory state that affects atherosclerotic disease, supporting the hypothesis that the presence of microorganisms regardless of the site of primary colonization can lead to general inflammatory changes that weigh on the progression of atherosclerosis.Finally, it could be possible that some bacteria can influence atherogenesis due to direct colonization of the vascular wall or simply because parts of it act as immune complexes capable of activating an inflammatory state responsible for the evolution of the atherosclerotic plaque [116].

Furthermore, an association between male gender and HP in carotid atherosclerotic plaques has also been highlighted, for which the predisposing factors would seem to take part both in determining gastric lesions and in the evolution of atherosclerosis, exploiting a predisposition of the host to develop gastric diseases or to have more marked atherosclerosis [117]. The link between inflammation and endothelial dysfunction has been taken into consideration by other studies, which have shown that acute phase proteins such as inflammatory adhesion molecules and C-reactive protein (CRP) were simultaneously elevated during HP infection and endothelial dysfunction [118]. Furthermore, chronic HP infection has been shown to correlate with levels of cytokines and clotting factors, which are in close relationship with the degeneration and instability of the atherosclerotic plaque [118]. The establishment of a hypercoagulable state associated with HP infection was the subject of another study [14] in which, starting from the assumption that TNF-α, produced following HP infection [119], causes the release of tissue factor from endothelial cells and monocytes by activating the extrinsic pathway of coagulation. The procoagulant activity of TNF-α has been measured, dosing plasma levels of the prothrombin fragment 1+2 which in turn leads to increased thrombin generation [120]. Consolazio et al. [14] demonstrated a significant increase in plasmatic TNF-α and F1+2 in HP-positive patients, compared to HP pylori-negative subjects, also observing how the eradication of the infection is associated with showing a diminution of all parameters, obtaining values observed in patients without HP infection. This result suggests that HP is an important inducer primarily of a conical inflammatory state and also of a state of hypercoagulability and therefore an added cardiovascular risk factor.

In conclusion, there is evidence that HP may contribute mildly as a cardiovascular risk factor to the progression of atherosclerosis, through the induction of low-grade inflammation and a procoagulant state. This contribution is particularly evident in male subjects, in the presence of other risk factors, and in the presence of gastritis during the activity phase. in particular, CagA-positive HP infection is associated with a mildly increased risk of CVD [121]. It would also appear that the eradication of HP achieves to some extent the decrease of the hypercoagulable state present in some patients. It still remains only in the hypotheses of how HP can lodge in atherosclerotic plaques or how it is able to influence endothelial function.

### 5.2. Experimental Evidence against the Possible Role of H. pylori in Atherosclerosis

The coexistence of HP infection with atherosclerosis has generated controversial findings on the causal relationship between the conditions over time (Figure 3). However, a causal relationship was disproved by findings indicating that commonly shared risk factors for HP gastritis and atherosclerosis, such as socioeconomic level with advanced age, cigarette smoking, and excess salt in the diet, can be confounding factors that deny their direct relationship [18].

In particular, unlike Ameriso et al. [116], Malnick et al. [122] did not detect the presence of HP in carotid endarterectomy samples taken from 10 male patients, not confirming the association between infection and atherosclerosis.

Blasi et al. [123,124] performed a similar study examining material derived from aortic aneurysms but without demonstration of HP traces. The negative results of these two works explain how difficult it is to find a link between the two conditions but also place the effectiveness of the investigative methods used in conflict. In fact, Ameriso et al. [116] used a polymerase chain reaction method that mostly amplified the gluM gene (ureC), which is very sensitive and specific for the detection of HP in gastric biopsies, compared to others [125]. The sensitivity of the former method would appear to be much higher than the latter.

Malnick et al. [122] used a species-specific antigenic method considered poorly sensitive, just as Blasi and colleagues [123,124] used the urease gene, which as a method does not achieve significant sensitivity. 

Akyön et al. [126] instead established a relationship between HP and atherosclerosis by detecting traces of HP but only in about 20% of the atherosclerotic plaques examined. Also, from the epidemiological point of view, some studies [127,128] have failed to find an association between HP and parameters of the progression of atherosclerosis.

## 6. Limits and Unanswered Questions

In recent years, we have seen efforts to demonstrate whether HP favors the evolution of atherosclerosis, but the reported results have been mixed and not always easily reproducible. HP infection is associated with a slightly increased risk of CVD, but being widespread, it should be sought in all patients at high cardiovascular risk [121]. A limitation in the analysis of the various results was the heterogeneity of the measured variables, circulating cytokines, inflammatory infiltrates, wall thickening, and haemo-coagulant activity which led to a compartmentalized view of the atherosclerotic phenomenon. A recent meta-analysis [129] showed that HP infection is associated with a significant increase in intima-media thickness and contrary to other evidence [109,110,111,112], the relationship was higher in younger and older adults, under 60, and in people with no history of cardiovascular events. So, considering the above, should we focus our prevention efforts on healthy subjects with evidence of HP infection, or on subjects with pre-existing cardiovascular risk factors and concurrent HP positivity and active gastritis (Cag-positive)?

In general, alongside the traditional determinants for the evolution of atherosclerosis, chronic inflammation plays an important role in the development of CVD according to a gradual and protracted mechanism. The relationship between inflammation and atherosclerosis has recently been supported by the involvement of innate immunity with the involvement of Toll-like receptors (TLRs) capable of influencing the host immune response. TLRs play a central role in promoting immune responses in chronic inflammatory processes such as atherosclerosis and HP gastritis and for this reason, future studies could be carried out to highlight this possible correlation.

## 7. Conclusions

In conclusion, at present, it has not yet been demonstrated whether atherosclerosis and HP-related gastritis are characterized by the same immune processes and whether there is a contextual relationship between them. Subsequent studies will have to be aimed at establishing a correlation between the immunogenic stimulus characterizing atherosclerosis and that characterizing HP infection.

## Figures and Tables

**Figure 1 biomedicines-12-00400-f001:**
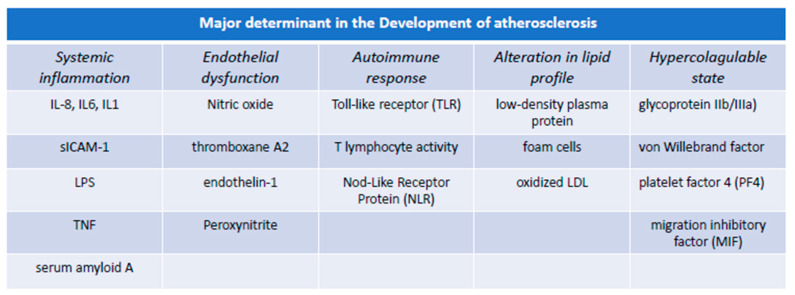
Major determinates in the atherosclerosis process.

**Figure 2 biomedicines-12-00400-f002:**
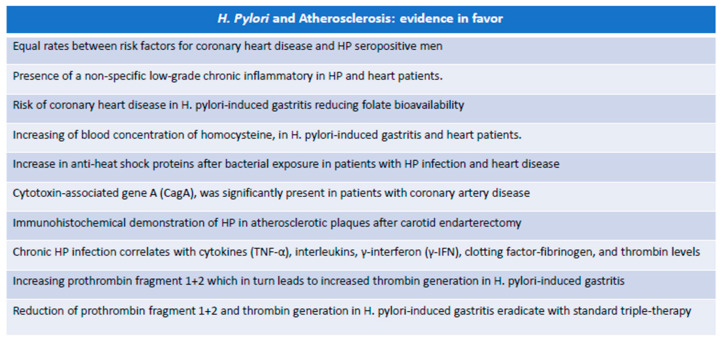
Evidence in favor of a possible role in *H. pylori* and atherosclerosis.

**Figure 3 biomedicines-12-00400-f003:**
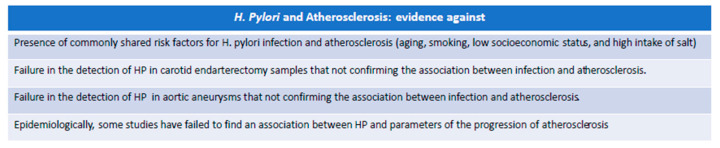
Evidence in favor of a possible role in H. pylori and Atherosclerosis.

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
