# Peer review of "Role of *Helicobacter pylori* Infection in Pathogenesis, Evolution, and Complication of Atherosclerotic Plaque"

_biomedicines, 2024, doi:10.3390/biomedicines12020400_

Round 1
Reviewer 1 Report
Comments and Suggestions for Authors
The manuscript was well revised. I have no further comments.
Author Response
Dear Reviewer 1
Thank you very much for approving our paper
Best regards
Reviewer 2 Report
Comments and Suggestions for Authors
I revise this manuscript after revisions. The authors have only partially responded to my comments. The paper is interesting, even if the argument is not innovative. I think the main critical point is the lack of figures and tables which would make the manuscript easier to read, even more so in an important journal such as Biomedicines. In my opinion the manuscript cannot be accepted in the present form
Comments on the Quality of English LanguageMinor revision
Author Response
Dear Reviewer 2
Thank you for your revision. We now describe our revision.
I revise this manuscript after revisions. The authors have only partially responded to my comments. The paper is interesting, even if the argument is not innovative. I think the main critical point is the lack of figures and tables which would make the manuscript easier to read, even more so in an important journal such as Biomedicines. In my opinion, the manuscript cannot be accepted in its present form
R: According to the suggestion of the reviewer, figures have been added to the paper
Comments on the Quality of English Language
Minor revision
R: According to the suggestion of the reviewer English language style has been revised
Thank you
Best regards
Reviewer 2
Comments and Suggestions for Authors
I revise this manuscript after revisions. The authors have only partially responded to my comments. The paper is interesting, even if the argument is not innovative. I think the main critical point is the lack of figures and tables which would make the manuscript easier to read, even more so in an important journal such as Biomedicines. In my opinion, the manuscript cannot be accepted in its present form
R: According to the suggestion of the reviewer, figures have been added to the paper
Comments on the Quality of English Language
Minor revision
R: According to the suggestion of the reviewer English language style has been revised
Reviewer 2
Comments and Suggestions for Authors
I revise this manuscript after revisions. The authors have only partially responded to my comments. The paper is interesting, even if the argument is not innovative. I think the main critical point is the lack of figures and tables which would make the manuscript easier to read, even more so in an important journal such as Biomedicines. In my opinion, the manuscript cannot be accepted in its present form
R: According to the suggestion of the reviewer, figures have been added to the paper
Comments on the Quality of English Language
Minor revision
R: According to the suggestion of the reviewer English language style has been revised
Reviewer 2
Comments and Suggestions for Authors
I revise this manuscript after revisions. The authors have only partially responded to my comments. The paper is interesting, even if the argument is not innovative. I think the main critical point is the lack of figures and tables which would make the manuscript easier to read, even more so in an important journal such as Biomedicines. In my opinion, the manuscript cannot be accepted in its present form
R: According to the suggestion of the reviewer, figures have been added to the paper
Comments on the Quality of English Language
Minor revision
R: According to the suggestion of the reviewer English language style has been revised
Reviewer 2
Comments and Suggestions for Authors
I revise this manuscript after revisions. The authors have only partially responded to my comments. The paper is interesting, even if the argument is not innovative. I think the main critical point is the lack of figures and tables which would make the manuscript easier to read, even more so in an important journal such as Biomedicines. In my opinion, the manuscript cannot be accepted in its present form
R: According to the suggestion of the reviewer, figures have been added to the paper
Comments on the Quality of English Language
Minor revision
R: According to the suggestion of the reviewer English language style has been revised
Reviewer 2
Comments and Suggestions for Authors
I revise this manuscript after revisions. The authors have only partially responded to my comments. The paper is interesting, even if the argument is not innovative. I think the main critical point is the lack of figures and tables which would make the manuscript easier to read, even more so in an important journal such as Biomedicines. In my opinion, the manuscript cannot be accepted in its present form
R: According to the suggestion of the reviewer, figures have been added to the paper
Comments on the Quality of English Language
Minor revision
R: According to the suggestion of the reviewer English language style has been revised
Reviewer 2
Comments and Suggestions for Authors
I revise this manuscript after revisions. The authors have only partially responded to my comments. The paper is interesting, even if the argument is not innovative. I think the main critical point is the lack of figures and tables which would make the manuscript easier to read, even more so in an important journal such as Biomedicines. In my opinion, the manuscript cannot be accepted in its present form
R: According to the suggestion of the reviewer, figures have been added to the paper
Comments on the Quality of English Language
Minor revision
R: According to the suggestion of the reviewer English language style has been revised
Reviewer 2
Comments and Suggestions for Authors
I revise this manuscript after revisions. The authors have only partially responded to my comments. The paper is interesting, even if the argument is not innovative. I think the main critical point is the lack of figures and tables which would make the manuscript easier to read, even more so in an important journal such as Biomedicines. In my opinion, the manuscript cannot be accepted in its present form
R: According to the suggestion of the reviewer, figures have been added to the paper
Comments on the Quality of English Language
Minor revision
R: According to the suggestion of the reviewer English language style has been revised
Reviewer 2
Comments and Suggestions for Authors
I revise this manuscript after revisions. The authors have only partially responded to my comments. The paper is interesting, even if the argument is not innovative. I think the main critical point is the lack of figures and tables which would make the manuscript easier to read, even more so in an important journal such as Biomedicines. In my opinion, the manuscript cannot be accepted in its present form
R: According to the suggestion of the reviewer, figures have been added to the paper
Comments on the Quality of English Language
Minor revision
R: According to the suggestion of the reviewer English language style has been revised
Reviewer 2
Comments and Suggestions for Authors
I revise this manuscript after revisions. The authors have only partially responded to my comments. The paper is interesting, even if the argument is not innovative. I think the main critical point is the lack of figures and tables which would make the manuscript easier to read, even more so in an important journal such as Biomedicines. In my opinion, the manuscript cannot be accepted in its present form
R: According to the suggestion of the reviewer, figures have been added to the paper
Comments on the Quality of English Language
Minor revision
R: According to the suggestion of the reviewer English language style has been revised
Reviewer 2
Comments and Suggestions for Authors
I revise this manuscript after revisions. The authors have only partially responded to my comments. The paper is interesting, even if the argument is not innovative. I think the main critical point is the lack of figures and tables which would make the manuscript easier to read, even more so in an important journal such as Biomedicines. In my opinion, the manuscript cannot be accepted in its present form
R: According to the suggestion of the reviewer, figures have been added to the paper
Comments on the Quality of English Language
Minor revision
R: According to the suggestion of the reviewer English language style has been revised
Reviewer 2
Comments and Suggestions for Authors
I revise this manuscript after revisions. The authors have only partially responded to my comments. The paper is interesting, even if the argument is not innovative. I think the main critical point is the lack of figures and tables which would make the manuscript easier to read, even more so in an important journal such as Biomedicines. In my opinion, the manuscript cannot be accepted in its present form
R: According to the suggestion of the reviewer, figures have been added to the paper
Comments on the Quality of English Language
Minor revision
R: According to the suggestion of the reviewer English language style has been revised
Reviewer 2
Comments and Suggestions for Authors
I revise this manuscript after revisions. The authors have only partially responded to my comments. The paper is interesting, even if the argument is not innovative. I think the main critical point is the lack of figures and tables which would make the manuscript easier to read, even more so in an important journal such as Biomedicines. In my opinion, the manuscript cannot be accepted in its present form
R: According to the suggestion of the reviewer, figures have been added to the paper
Comments on the Quality of English Language
Minor revision
R: According to the suggestion of the reviewer English language style has been revised
Reviewer 2
Comments and Suggestions for Authors
I revise this manuscript after revisions. The authors have only partially responded to my comments. The paper is interesting, even if the argument is not innovative. I think the main critical point is the lack of figures and tables which would make the manuscript easier to read, even more so in an important journal such as Biomedicines. In my opinion, the manuscript cannot be accepted in its present form
R: According to the suggestion of the reviewer, figures have been added to the paper
Comments on the Quality of English Language
Minor revision
R: According to the suggestion of the reviewer English language style has been revised
Reviewer 2
Comments and Suggestions for Authors
I revise this manuscript after revisions. The authors have only partially responded to my comments. The paper is interesting, even if the argument is not innovative. I think the main critical point is the lack of figures and tables which would make the manuscript easier to read, even more so in an important journal such as Biomedicines. In my opinion, the manuscript cannot be accepted in its present form
R: According to the suggestion of the reviewer, figures have been added to the paper
Comments on the Quality of English Language
Minor revision
R: According to the suggestion of the reviewer English language style has been revised
Reviewer 2
Comments and Suggestions for Authors
I revise this manuscript after revisions. The authors have only partially responded to my comments. The paper is interesting, even if the argument is not innovative. I think the main critical point is the lack of figures and tables which would make the manuscript easier to read, even more so in an important journal such as Biomedicines. In my opinion, the manuscript cannot be accepted in its present form
R: According to the suggestion of the reviewer, figures have been added to the paper
Comments on the Quality of English Language
Minor revision
R: According to the suggestion of the reviewer English language style has been revised
Reviewer 2
Comments and Suggestions for Authors
I revise this manuscript after revisions. The authors have only partially responded to my comments. The paper is interesting, even if the argument is not innovative. I think the main critical point is the lack of figures and tables which would make the manuscript easier to read, even more so in an important journal such as Biomedicines. In my opinion, the manuscript cannot be accepted in its present form
R: According to the suggestion of the reviewer, figures have been added to the paper
Comments on the Quality of English Language
Minor revision
R: According to the suggestion of the reviewer English language style has been revised
Reviewer 3 Report
Comments and Suggestions for Authors
1. Lines 57-62: "Nitric oxide (NO) is an important determinant of systemic circulation and actively contributes to modulating processes such as inhibition of platelet adhesion and aggregation, maintenance of the physiological base pressure of the arteries (8), being an obstacle to the development of atherosclerotic plaque (9) and minimize the proliferation and increase in the thickness of vascular smooth muscle (10)." The second part of this sentence is not only grammatically incorrect, but also unclear.
2. The objective of the manuscript should be clearly mentioned at the end of the introduction chapter.
3. Why did you choose to include meta-analysis in this narrative review?
4. Line 93: "data extraction and quality assessment of the papers were conducted by two reviewers (PC and TC) who independently extracted the following data...". Which following data?
5. Chapter 4.1. can be shortened, as it deviates from the review's objectives.
6. The authors mention "Inclusion criteria were as follows: clinical, molecular, experimental, and observational studies between HP positive and HP negative patients, metaanalysis." Yet, they mainly report results of experimental data
7. Conclusions should be original, include a personal closure of the present review and not include any reference to previous work.
8. The article could benefit from a table which summarizes current studies which sustain the involvement of H. pylori in the etiopathogenesis of atherosclerosis, as well as a figure which highlights the underlying mechanisms which link H. pylori to atherosclerosis.
9. Minor English corrections are required.
Comments on the Quality of English LanguageMinor English corrections are required.
Author Response
Reviewer 3
Comments and Suggestions for Authors
- Lines 57-62: "Nitric oxide (NO) is an important determinant of systemic circulation and actively contributes to modulating processes such as inhibition of platelet adhesion and aggregation, maintenance of the physiological base pressure of the arteries (8), being an obstacle to the development of atherosclerotic plaque (9) and minimize the proliferation and increase in the thickness of vascular smooth muscle (10)." The second part of this sentence is not only grammatically incorrect but also unclear.
R: According to the suggestion of the reviewer sentence has been revised and corrected
- The objective of the manuscript should be clearly mentioned at the end of the introduction chapter.
- Why did you choose to include a meta-analysis in this narrative review?
R: According to the suggestion of the reviewer, there is a mistake in the sentences present in the section “methods”
- Line 93: "data extraction and quality assessment of the papers were conducted by two reviewers (PC and TC) who independently extracted the following data...". Which following data?
R: According to the suggestion of the reviewer sentence has been revised
- Chapter 4.1. can be shortened, as it deviates from the review's objectives.
R: In accordance with the reviewer's suggestions, chapter 4 of the manuscript has been further summarized and reduced to its extent.
- 6. The authors mention "Inclusion criteria were as follows: clinical, molecular, experimental, and observational studies between HP positive and HP negative patients, meta-analysis." Yet, they mainly report the results of experimental data
R: According to the suggestion of the reviewer, the term meta-analysis has been removed from the text
- Conclusions should be original, include a personal closure of the present review, and not include any reference to previous work.
R: In accordance with the reviewer's suggestions, the chapter of the conclusions was separated by creating two chapters: Chapter 6 is dedicated to the limits of the research and unanswered questions; Chapter 8 briefly summarizes the conclusions of the research.
- The article could benefit from a table that summarizes current studies that sustain the involvement of H. pylori in the etiopathogenesis of atherosclerosis, as well as a figure that highlights the underlying mechanisms that link H. pylori to atherosclerosis.
R: According to the suggestion of the reviewer, figures have been added to the paper
- Minor English corrections are required.
R: According to the suggestion of the reviewer English language style has been revised
Comments on the Quality of English Language
Minor English corrections are required.
R: According to the suggestion of the reviewer English language style has been revised
Reviewer 3
Comments and Suggestions for Authors
- Lines 57-62: "Nitric oxide (NO) is an important determinant of systemic circulation and actively contributes to modulating processes such as inhibition of platelet adhesion and aggregation, maintenance of the physiological base pressure of the arteries (8), being an obstacle to the development of atherosclerotic plaque (9) and minimize the proliferation and increase in the thickness of vascular smooth muscle (10)." The second part of this sentence is not only grammatically incorrect but also unclear.
R: According to the suggestion of the reviewer sentence has been revised and corrected
- The objective of the manuscript should be clearly mentioned at the end of the introduction chapter.
- Why did you choose to include a meta-analysis in this narrative review?
R: According to the suggestion of the reviewer, there is a mistake in the sentences present in the section “methods”
- Line 93: "data extraction and quality assessment of the papers were conducted by two reviewers (PC and TC) who independently extracted the following data...". Which following data?
R: According to the suggestion of the reviewer sentence has been revised
- Chapter 4.1. can be shortened, as it deviates from the review's objectives.
R: In accordance with the reviewer's suggestions, chapter 4 of the manuscript has been further summarized and reduced to its extent.
- 6. The authors mention "Inclusion criteria were as follows: clinical, molecular, experimental, and observational studies between HP positive and HP negative patients, meta-analysis." Yet, they mainly report the results of experimental data
R: According to the suggestion of the reviewer, the term meta-analysis has been removed from the text
- Conclusions should be original, include a personal closure of the present review, and not include any reference to previous work.
R: In accordance with the reviewer's suggestions, the chapter of the conclusions was separated by creating two chapters: Chapter 6 is dedicated to the limits of the research and unanswered questions; Chapter 8 briefly summarizes the conclusions of the research.
- The article could benefit from a table that summarizes current studies that sustain the involvement of H. pylori in the etiopathogenesis of atherosclerosis, as well as a figure that highlights the underlying mechanisms that link H. pylori to atherosclerosis.
R: According to the suggestion of the reviewer, figures have been added to the paper
- Minor English corrections are required.
R: According to the suggestion of the reviewer English language style has been revised
Comments on the Quality of English Language
Minor English corrections are required.
R: According to the suggestion of the reviewer English language style has been revised
Thank you
Best regards
Round 2
Reviewer 3 Report
Comments and Suggestions for Authors
1. Lines 77-81: please rewrite the paragraph stating the objective of the manuscript, as the adjective “useful” and the noun “usefulness” are repeated for a total number of three times among the same phrase.
2. Comment 6 has not been addressed/clarified.
Comments on the Quality of English LanguageMinor English language revision is required.
Author Response
Reviewer 2
- Lines 77-81: Please rewrite the paragraph stating the objective of the manuscript, as the adjective “useful” and the noun “usefulness” are repeated for a total number of three times among the same phrase.
R: According to the suggestion of the reviewer, the sentences have been modified
- Comment 6 has not been addressed/clarified.
R: According to the suggestion of the reviewer, an attempt has been made to better clarify the contents of comment number 6, by modifying the sentence.
Comments on the Quality of English Language
Minor English language revision is required.
R: According to the suggestion of the reviewer, the English language revision will be finalized in the final proofreading of the manuscript
Reviewer 2
- Lines 77-81: Please rewrite the paragraph stating the objective of the manuscript, as the adjective “useful” and the noun “usefulness” are repeated for a total number of three times among the same phrase.
R: According to the suggestion of the reviewer, the sentences have been modified
- Comment 6 has not been addressed/clarified.
R: According to the suggestion of the reviewer, an attempt has been made to better clarify the contents of comment number 6, by modifying the sentence.
Comments on the Quality of English Language
Minor English language revision is required.
R: According to the suggestion of the reviewer, the English language revision will be finalized in the final proofreading of the manuscript
Reviewer 2
- Lines 77-81: Please rewrite the paragraph stating the objective of the manuscript, as the adjective “useful” and the noun “usefulness” are repeated for a total number of three times among the same phrase.
R: According to the suggestion of the reviewer, the sentences have been modified
- Comment 6 has not been addressed/clarified.
R: According to the suggestion of the reviewer, an attempt has been made to better clarify the contents of comment number 6, by modifying the sentence.
Comments on the Quality of English Language
Minor English language revision is required.
R: According to the suggestion of the reviewer, the English language revision will be finalized in the final proofreading of the manuscript